# Short-Term High-Intensity Circuit Training Does Not Modify Resting Heart Rate Variability in Adults during the COVID-19 Confinement

**DOI:** 10.3390/ijerph19127367

**Published:** 2022-06-16

**Authors:** Patricia C. García-Suárez, Jorge A. Aburto-Corona, Iván Rentería, Luis M. Gómez-Miranda, José Moncada-Jiménez, Fábio Santos Lira, Barbara Moura Antunes, Alberto Jiménez-Maldonado

**Affiliations:** 1Facultad de Deportes Ensenada, Universidad Autónoma de Baja California, Ensenada 22890, Mexico; patricia.garcia@uabc.edu.mx (P.C.G.-S.); irenteria@uabc.edu.mx (I.R.); barbara.moura@uabc.edu.mx (B.M.A.); 2Department of Health, Sports and Exercise Sciences, University of Kansas, Lawrence, KS 66045, USA; 3Facultad de Deportes Tijuana, Universidad Autónoma de Baja California, Tijuana 22615, Mexico; jorge.aburto@uabc.edu.mx (J.A.A.-C.); luismariouabc@gmail.com (L.M.G.-M.); 4Human Movement Sciences Research Center (CIMOHU), University of Costa Rica, San José 1200, Costa Rica; jose.moncada@ucr.ac.cr; 5Exercise and Immunometabolism Research Group, Department of Physical Education, Paulista State University (UNESP), Presidente Prudente 19060-900, Brazil; fabio.lira@unesp.br

**Keywords:** COVID-19 quarantine, heart rate variability, high-intensity circuit training, stress

## Abstract

Background/Objective: The quarantine caused by the COVID-19 pandemic increased sedentary behavior, psychological stress, and sleep disturbances in the population favoring the installation of alterations in the cardiovascular system. In this sense, physical exercise has widely been suggested as an efficient treatment to improve health. The current study determined the impact of short-term high-intensity circuit training (HICT) on resting heart rate variability (HRV) in adults. Methods: Nine healthy participants (age: 31.9 ± 4.4 yr.) performed 36 HICT sessions (3 times per day; 3 days per week) and four participants (age: 29.5 ± 1.7 yr.) were assigned to a control group. The HICT consisted of 12 min of whole-body exercises performed during a workout. Twenty-four hours before and after the exercise program, HRV parameters were recorded. Results: The heart rate exercise during the last session trended to be lower when compared with the first HICT session (*p* = 0.07, d = 0.39, 95% CI = −13.50, 0.72). The interval training did not modify the HRV time (Mean NN, SDNN, RMSSD, NN50, pNN50) and frequency (LF, HF, LF/HF ratio, total power) domain parameters. Conclusion: Thirty-six HICT sessions did not provide enough stimuli to modify the resting HRV in adults during social isolation elicited by the COVID-19 pandemic. However, the data suggested that exercise protocol did not induce cardio-vagal adaptations.

## 1. Introduction

The Severe Acute Respiratory Syndrome Coronavirus 2 (SARS-CoV-2) forced government authorities of several countries, such as Mexico [1], to impose strict cautionary strategies of physical distancing to prevent the spread of the infection due to the sudden growth of COVID-19 cases worldwide and the gravity of the disease. As a result of this action, more than 4 billion inhabitants were confined to their homes worldwide, which affected people’s job routines and daily physical activity, leading to an increase in physical inactivity, sedentary behavior, and its known deleterious consequences [2,3,4]. It is widely known that sedentary behavior causes body weight gain, harms cardiovascular function, impairs the immune system response, and increases the risk of suffering physiological disorders (e.g., depression, psychological stress, and anxiety) [2,3,4]. Furthermore, evidence from the earlier stages of the COVID-19 expansion identified vulnerable groups to SARS-CoV-2, such as individuals with cardiometabolic disease, obesity, and aging (i.e., frailty) [5]. Sedentary behavior during lockdown exacerbated these medical conditions [3].

The World Health Organization (WHO) considers physically active to be an adult that reaches at least 150 min of weekly moderate- or 75 min vigorous-intensity activity or an equivalent combination of both [6]; nevertheless, during home confinement, it is very challenging to accomplish these physical activity recommendations. Recently, Narici et al. indicated a lack of consensus regarding the minimum dose and frequency of physical activity during lockdown to protect people’s health from inactivity [7]. Therefore, exercise scientists have focused on studying the impact of time-efficient short exercise bouts on human physiology to strengthen health [8].

High-intensity interval training (HIIT) is an exercise modality characterized by short bursts of maximal or submaximal exertion (e.g., ≥80–95% of the heart rate maximum (HRmax)), interspersed with recovery periods at ≤50% of HRmax, requiring ≤30 min to complete an exercise session [8]. Although HIIT can be performed in different exercise modalities (aerobic, anaerobic, circuit), high-intensity circuit training (HICT) incorporates multi-stimulating, circuit-like, multiple-joint exercises and uses body weight as resistance [9]. Therefore, HICT could be an effective workout in a home exercise setting because it does not require equipment and can be performed in reduced spaces only using a person’s own body weight [10,11].

Before the COVID-19 pandemic, studies showed that reducing sedentary behavior had measurable positive effects on heart rate variability (HRV) [12,13,14,15,16,17]. HRV is a reliable technique to measure the cardiac autonomic tone and its association with the brain–heart interaction produced by the vagus and sympathetic nerves [9]. Furthermore, HRV is a highly sensitive indicator of physiological effects [12], psychological distress [13,14], and cardiovascular disease risk factors [15]. Thus, it is a practical tool for measuring the physical and mental status of the general population [16,17]. The effect of HICT on the HRV was previously studied [18,19]; however, the participants in these studies followed a free daily lifestyle (i.e., they did not suffer social isolation and leisure activity restrictions due to a pandemic situation). Therefore, the purpose of the present study was to assess the impact of a short-term HICT intervention on HRV in healthy adults during the COVID-19 quarantine. In this study, we hypothesized that home-based HICT intervention would induce positive effects on the HRV. The novelty of this research is that home-based exercise could be an effective stimulus for maintaining physiological health during the COVID-19 confinement, as was suggested [20,21].

## 2. Materials and Methods

### 2.1. Participants

Eighteen adult employees from the Facultad de Deportes at Universidad Autónoma de Baja California (age: 34.2 ± 5.4 yr.) were recruited for the study. The inclusion criteria were that volunteers had no physical injuries or limitations which would prevent them from performing high-intensity physical activity, additionally, the researchers only included those volunteers who felt comfortable participating in the study despite the prolonged lockdown side effects. The exclusion criterion was that throughout the study any participant or a family member living at home tested positive for COVID-19. Participants were randomly assigned to HICT (n = 9) and control (-CON-n = 9) by simple randomization on an Excel spreadsheet. Five participants withdrew from the study (Figure 1); therefore, 13 participants (age: 31.1 ± 3.8 yr.) completed the entire study (HICT: 9 m/f: 4/5, CON: 4 m/f, 0/4), and were included in the final analysis.

### 2.2. Experimental Procedures

The recruitment process consisted of several steps. First, the research team sent electronic invitations to all workers from the “Facultad de Deportes” at “Universidad Autónoma de Baja California” to participate in the study. Second, those who replied to the invitation were appointed to attend a virtual meeting where members of the research team explained in detail the aim and procedures of the study. After this, all the participants agreeing to participate in the study read and signed a written informed consent set up in Google Forms. Before and after HICT, the participants completed an electronic version of the short-form International Physical Activity Questionnaire (IPAQ-SF) to subjectively assess their physical activity levels. The IPAQ-SF consists of seven questions asking individuals to recall the previous week’s physical activity and asks them about the number of days and the amount of time spent walking, sitting, or participating in moderate- and vigorous-intensity activities [22]. Additionally, the Physical Activity Readiness Questionnaire (PAR-Q) was applied in order to assess the participants’ health status to perform highly demanding exercises via Google Forms [23]. Then, the researchers personally delivered one chest heart rate monitor Polar H10 (Polar Electro Oy, Kempele, Finland) to each participant. The guidelines of the World Health Organization (WHO) to prevent the spread of COVID-19 were rigorously followed. Finally, a videotape was filmed by qualified fitness trainers to demonstrate the HICT sessions to the participants in the exercise (HICT) group. Participants in the control group (CON) did not HICT but were allowed to practice physical exercise in concordance with their lifestyle during the lockdown. All procedures were approved by the Research Ethics Committee of Facultad de Medicina y Psicología Campus Tijuana and the protocol was registered under the code 889/2020-2.

### 2.3. Heart Rate Variability Assessment

Before the HRV collection, all participants were asked to avoid drinking caffeine and alcohol and to refrain from strenuous physical exercise for at least 12 h before evaluation. For the morning (8:00–9:00 a.m.) HRV measurement, the participants were instructed to remain in a supine position for 10 min before the beat-to-beat intervals (R–R) were registered using the Elite HRV^®^ app (Elite HRV LLC, Asheville, NC, USA, and Release 4.0.2, 2018). The first five minutes of the recording were ignored to avoid noise data associated with involuntary movements and artifact correction was set at very low.

Then, the participants exported the data and emailed it to the researchers, who analyzed it with the Kubios Heart Rate Variability standard version software (Kubios HRV 3.4.2, Oy, Finland). The beat correction was set at very low to avoid possible noise data associated with involuntary movements. The time–domain parameters measured were mean heart rate (HR), mean R–R, the standard deviation of beat-to-beat (SDNN), root mean square of the successive differences (RMSSD), direct and relative successive beats with >50 ms of difference (NN50 and pNN50). Finally, the frequency domains measured were low-frequency (LF), high-frequency (HF) in normal units (n.u.), LF/HF ratio, and total power. Frequency cut-off values were set at 0.05–0.15 for LF and 0.16–0.4 for HF with the Fast Fourier Transformation method (FFT).

### 2.4. High-Intensity Circuit Training

The HICT program lasted one month, for a total of 36 exercise sessions. The program was delivered in May 2020 and it was performed three times per day on Monday, Wednesday, and Friday. The exercise sessions were completed in the mornings (7:00–9:00 a.m.), afternoons (1:00–3:00 p.m.), and evenings (6:00–9:00 p.m.). The participants wore a heart rate monitor and recorded the data on the Elite HRV^®^ app during the exercise sessions. The session length was 12 min, including rest and workout periods. In detail, each exercise session started with a warm-up of 2 min, which included stretching exercises aimed at improving the joint range of motion. Immediately after warm-up, participants started a modified version of the program described by Schleppenbach et al. [24]. The program comprised workout exercises during 30 s: jumps, high knees, line jumps, burpees, and push-ups. Each exercise was followed by 30 s passive recovery [24]. The circuit was performed twice to complete 10 min.

### 2.5. Statistical Analysis

The statistical analyses were performed with the GraphPad Prism software, version 6.0 (GraphPad Inc., San Diego, CA, USA, Release 6.01, 2012) and the R software version 4.1.1 (R Core Team) [25] in the RStudio environment version 1.4.1717 (RStudio Team) [26]. Single group session analysis for the HICT group was conducted with a dependent Student’s t-test, the *p*-value was set at ≥0.05 for any significant change. Effect size was calculated by Cohen’s d (d) and analyzed as proposed by Cohen (small: <0.2–0.49; moderate: 0.5–0.79; and large: >0.8) [27]. The summary data are reported as the mean and standard deviation (±SD). The inferential analysis included an analysis of covariance (ANCOVA) that removed the initial error variance in both control and experimental groups. Each data obtained after the training period was taken and used as a response variable to form a regression using as a design factor the subjects’ grouping variable (i.e., control, exercise) and, as a covariate, the measurements before starting the experimental period. The regression model built used the concept of the sum of squares of marginal regression to analyze the variability that explains whether performing HICT or not once the corresponding covariate was included, and that has previously explained a part of the total sum of squares. After obtaining regressions for each HRV variable, ANOVA tests of each model were performed to determine if the assumption that the covariate is not associated with the treatment is met. Thus, we computed 13 models for the HRV variables. First, the model analysis including the interaction was studied; however, statistical significance was not reached (i.e., *p* > 0.05); therefore, the interaction was not considered for the models used, which allowed us to verify the assumption of independence between the covariate and the treatment.

Finally, an analysis of the individual responses was performed by obtaining the standard deviation of the difference scores and the within-subject standard deviation (WSSD), the technical error of measurement (TEM) and the smallest worthwhile change (SWC threshold) using the formula SWC = 0.5 × TEM [28].

## 3. Results

Anthropometric data and baseline estimated self-reported energy expenditure (METs) are presented in Table 1, classifying them as moderately active individuals. For the HICT group, the HR response at every high-intensity bout during the first session (S1) and the last session (S36) is shown in Figure 2A. The mean HR at S1 was 143 ± 16 bmp and at S36 was 136 ± 18 bmp (*p* = 0.07, d = 0.39, 95% CI = −13.50, 0.72). The time spent in vigorous physical activity for the HICT group is shown in Figure 2B. At the beginning of the program, the time spent in vigorous physical activity was 27.5 ± 19.6 min per day (min/day), and at the end of the program was 47.5 ± 24.20 min/day (*p* = 0.005, d = 0.69, 95% CI = 8.177, 31.82).

The inferential analysis is presented in Table 2. With a significance level of 5%, no mean differences were found in each subject for the factor levels (i.e., control or HICT) on each of the 13 HRV variables. Furthermore, for each HRV variable, the mean obtained after the intervention was not different regardless of whether they belonged to the control group or the exercise group. Table 2 also shows the power of the test, which corresponds to the probability of detecting a difference equal to or greater than the threshold recorded as the relevant difference between the control and the HICT groups for the respective response variable. Again, very high-power values close to 1 were found, which indicates a very low probability of not rejecting the null hypothesis given that it is false, which is relevant due to the results obtained in the ANOVA tests. These results allow us to conclude that the absence of significance is not due to the sample size.

Individual responses to four selected HRV parameters for all individuals are presented in Figure 3. The upper and lower thresholds are depicted by continuous lines on the Y-axis of each figure. Individuals located between those lines are considered non-responders on that specific HRV parameter. For instance, Figure 3A shows that for the Δmean RR, three individuals (33%) in the HICT group (black bar) and one individual (25%) in the control group (grey bar) responded to the intervention. Figure 3B shows that for the ΔRMSSD parameter, only one individual (11%) in the HICT group (black bar) responded to the intervention. For the ΔLF two individuals (22%) in the HICT group (black bar) and two individuals (50%) in the control group (grey bar) responded to the intervention, and for the ΔHF domain parameters, five individuals (56%) in the HICT group (black bar) and one individual (25%) in the control group (grey bar) responded to the intervention (Figure 3C,D).

## 4. Discussion

The current work assessed the impact of a short-term HICT program on HRV parameters during the COVID-19 lockdown in adults. The data of the current study demonstrated that one month of lockdown during COVID-19 did not modify the resting HRV in adults.

In agreement with others [29], the energy expenditure data showed vigorous physical activity engagement at baseline [22] (Table 1). Additionally, the IPAQ data indicated that all participants maintained high leisure activities during the third month of the lockdown; these results are in agreement with recent reports [30,31]. This finding is relevant for population health. Furthermore, recent cross-sectional studies indicate that during the COVID-19 confinement, a higher physical activity level is associated with better physical and mental health [32,33].

A reduction in resting HR was found following the program compared with the values recorded at the beginning of the intervention. A reduction in the HR during the exercise is considered a physiological marker associated with an improvement in fitness [34,35]. These data reinforce a positive effect of 36 HICT sessions on the fitness level of the participants. Little et al. have demonstrated that sprint snacks involving 3 × 20 s ‘all out’ cycling bouts separated by 1–4 h rest, improved cardiorespiratory fitness after 6 weeks in healthy young, inactive adults [36]. Recently, a study by Caldwell et al. enrolled healthy males in a period of 6–8.5 h of sitting, and an hourly staircase sprint interval exercise (approx. 14–20 s each) [37]. Sprint snacks improved femoral artery blood flow and shear patterns following prolonged sitting. Taken together, our data demonstrated that high-intensity exercise for a short time can be an excellent strategy to maintain cardiorespiratory fitness.

For autonomic control of the heart, the HRV was not modified by the 36 sessions of HICT. Comparatively, other studies have reported null effects of functional training on HRV [18,19]. In addition, Lira et al. did not find a change in HR kinetics during the first and last (5 weeks, 3 times per week, both ran 5 km on a treadmill) exercise session of the high-intensity interval training or moderate-intensity continuous training in young males [38].

Contrarily, other authors reported an improvement in the HRV after eight sessions of interval training with cycling exercises [39]. In the same sense, an increase in the spectral high-frequency component, R-R interval, and a decrease in the spectral low-frequency component (nu) was observed after 8 weeks of running interval training plus strength training [40]. These findings highlight the exercise modality to induce changes in HRV in adults. One of the possible reasons can be differences in physiological demands observed among interval training using only leg exercises versus HIIT which requires a variety of movements and muscle groups [11,41], and the volume of intermittent exercise sessions induces a deep impact on HRV [42].

The participants of the current study reported IPAQ values considered moderately active (Table 1). Several reports have demonstrated that physical activity level influences HRV adaptations from exercise training [43,44]. The present results showed no significant changes in spectral HRV analyses (e.g., H.F., L.F., LF/HF) as these could have been found in athletes and physically active populations during a short-term intervention [43,44].

Bechke et al. studied a group of women and did not report changes in HRV following 16 weeks of high-intensity functional training [19]. The researchers did not control the exercise programming and the intervention compliance of the sample [19]. Regarding this, it was shown that there is a gender-specific response to the sympathovagal modulation [45]. Specifically, women tend to have a greater vagal signal in HRV compared to men [46]. The final sample of our study was comprised mostly of women (HICT: 5, CON: 4), another rationale for the higher parasympathetic activity outcome at baseline in both groups.

In line with this, the global HRV observed through the SDNN variable was not different between groups, also not showing any change after 4 weeks of HICT, remaining in an average SDNN value [17]. The time–domain marker for parasympathetic activity (i.e., RMSSD) was not affected in both groups, as well as the marginal values for surveying the vagal tone from HR and RR [17], showing that HICT does not alter the parasympathetic modulation in physically active adults under pandemic lockdown.

The main finding of the current study was the lack of effect of a home-based HICT intervention on HRV markers in an adult sample. Even though there was no improvement between groups, it demonstrates that HICT is an exercise modality that does not induce overtraining (not disruptions in LF/HF and RMSSD indexes [47])) in a short-volume, high-session per week demand, and therefore can be considered an aid to reduce the sedentary lifestyle induced by the COVID-19 lockdown.

The current study shows several limitations that can be addressed by future studies to elucidate the impact of the HICT on adults during a long period of lockdown. We did not apply physical fitness tests to assess the impact of HICT on fitness levels (e.g., aerobic capacity, muscular endurance) in the adult population during confinement. However, the HR during the exercise (Table 2) suggests a positive impact of interval training on aerobic capacity. Another limitation is the lack of control over the diet pattern of the participants. This is relevant because others have indicated that micro and macronutrient intake could regulate HRV; thus, future studies are needed to elucidate the impact of diet patterns and exercise on HRV in adults during confinement. Equally, we did not control or regulate the menstrual cycle in female participants, even though there is some trending evidence in regard to menstrual cycle fluctuations with HRV, the timely execution of the intervention was crucial due to the incertitude of the confinement period established by the Mexican health authorities. Moreover, the current study showed an unequal sample size for each group due to participant withdrawal; however, the post-hoc power analysis showed that the findings were not influenced by the sample size (Table 2). Finally, further studies would be needed to evaluate the impact of physical training on the subject’s engagement and the evaluation of metabolic markers associated with sedentary behavior in the adult population during the pandemic lockdown.

## 5. Conclusions

In conclusion, this study showed that HICT sessions did not provide enough stimuli to modify the resting HRV in adults during social isolation elicited by the COVID-19 pandemic. However, the data suggested that an exercise protocol did not induce stress.

## Figures and Tables

**Figure 1 ijerph-19-07367-f001:**
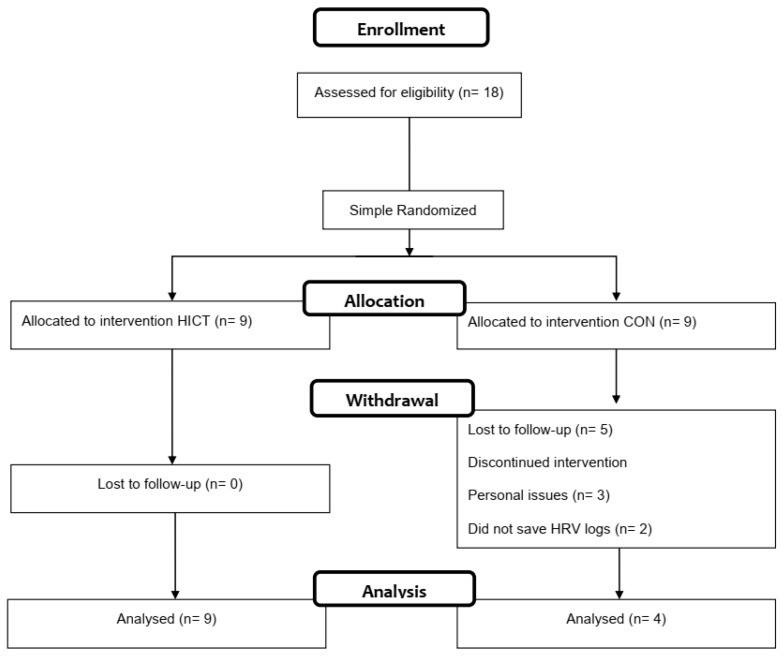
Flow chart of the intervention protocol.

**Figure 2 ijerph-19-07367-f002:**
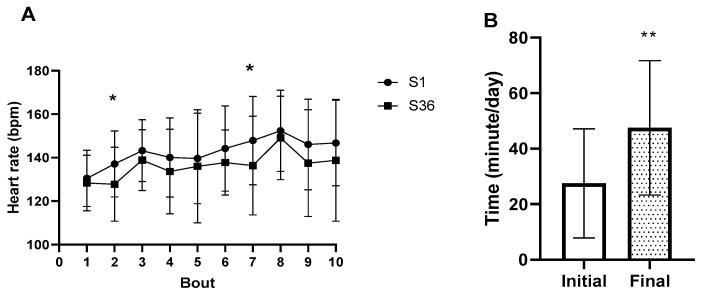
Heart rate response during first HICT session (S1) and last HICT session (S36) (**A**). Self-reported time spent with vigorous physical activity for HICT group (**B**). Student’s *t*-test, * *p* < 0.05, ** *p* < 0.01.

**Figure 3 ijerph-19-07367-f003:**
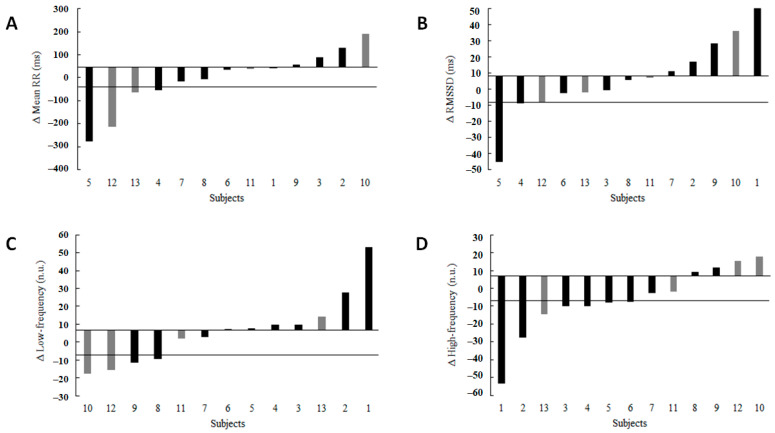
Individual responses of all participants for selected HRV parameters. Black bars belong to the HICT, while gray bars belong to CON. The upper and lower thresholds are depicted by continuous lines on the Y-axis of each figure. Individuals located between those lines are considered non-responders on that specific HRV parameter. (**A**) shows that three individuals in the HICT group and one individual in the control group responded to the intervention. (**B**) shows that only one individual in the HICT group responded to the intervention. (**C**) shows that two individuals in the HICT group and two individuals in the control group responded to the intervention. (**D**) shows that five individuals in the HICT group and one individual in the control group responded to the intervention.

**Table 1 ijerph-19-07367-t001:** Descriptive data of the sample.

Variable	CON (n = 4) m/f (0/4)	HICT (n = 9) m/f (4/5)	*p*≤
Age (yr.)	29.5 ± 1.7	31.9 ± 4.4	0.320
Weight (kg)	68.6 ± 6.9	80 ± 18.1	0.250
Height (cm)	160.0 ± 4.0	167.0 ± 8.0	0.001
BMI (kg/m^2^)	26.39 ± 2.6	28.47 ± 4.8	0.440
IPAQ (METs) *	2253.25 ± 1933	1240.81 ± 1104	0.260
Vigorous activity (min/day)	7.5 ± 15.0	27.5 ± 19.6	0.106

Data are presented as mean ± SD. Student’s *t*-test. BMI: Body mass index. IPAQ: International physical activity questionnaire. * Sedentary < 600, Moderate < 3000, and Vigorous > 3000 METs. m/f: Male to female distribution.

**Table 2 ijerph-19-07367-t002:** Probability and test power for each resting heart rate variability parameter. Under the analysis of covariance relevant difference.

Variable	Group	Mean ± sd	*p*=	Cohen’s d	Power (1 − ß)	Smallest Worthwhile Change (Threshold)
Mean RR (ms)	*CON*	910.0 ± 113.9	0.796	−0.248	0.99	45.3
	*HICT*	965.3 ± 251.7				
SDNN (ms)	*CON*	50.4 ± 19.3	0.587	−0.368	0.99	5.5
	*HICT*	59.2 ± 25.3				
Mean HR (bpm)	*CON*	66.7 ± 8.6	0.710	−0.033	1.00	6.3
	*HICT*	67.4 ± 23.5				
SDHR (bpm)	*CON*	3.8 ± 1.2	0.642	−0.218	0.99	1.4
	*HICT*	4.6 ± 4.5				
Min HR (bpm)	*CON*	58.4 ± 5.1	0.236	0.225	1.00	3.0
	*HICT*	55.8 ± 13.1				
Max HR (bpm)	*CON*	82.6 ± 12.3	0.593	0.053	0.99	7.6
	*HICT*	81.4 ± 25.2				
RMSSD (ms)	*CON*	47.2 ± 28.4	0.983	−0.552	0.98	8.4
	*HICT*	66.5 ± 37.2				
NN50 (beats)	*CON*	71.3 ± 67.0	0.375	−0.212	1.00	20.4
	*HICT*	85.6 ± 67.6				
pNN50 (%)	*CON*	23.3 ± 24.0	0.109	−0.318	1.00	8.4
	*HICT*	31.4 ± 25.7				
LF (n.u.)	*CON*	58.7 ± 23.6	0.205	0.136	1.00	6.7
	*HICT*	55.6 ± 22.7				
HF (n.u.)	*CON*	41.2 ± 23.6	0.204	−0.136	1.00	6.7
	*HICT*	44.3 ± 22.7				
Total power (ms^2^)	*CON*	2657.2 ± 1837.3	0.787	−0.487	1.00	935.1
	*HICT*	7506.1 ± 11,630.5				
LF/HF	*CON*	2.2 ± 1.9	0.623	−0.053	1.00	1.3
	*HICT*	2.3 ± 2.5				

Abbreviations: Mean RR—Mean inter-beat intervals length, SDNN—the standard deviation of beat-to-beat, Mean HR—Mean heart rate, SDHR—standard deviation of heart rate, Min HR—minimum heart rate, Max HR—maximal heart rate, RMSSD—root mean square of the successive differences, NN50—total successive beats with >50 ms of difference, pNN50—relative successive beats with >50 ms of difference, LF—Low-Frequency domain, HF—High-Frequency domain, LF/HF—Low-to-High Frequency domain ratio, ms—milliseconds, bpm—beats per minute, n.u.—normal units.

## Data Availability

The data used and/or analyzed during the study are available from the corresponding author on reasonable request.

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
