# Peer review of "Short-Term High-Intensity Circuit Training Does Not Modify Resting Heart Rate Variability in Adults during the COVID-19 Confinement"

_ijerph, 2022, doi:10.3390/ijerph19127367_

Round 1
Reviewer 1 Report
- The first question for the authors refers to the number of subjects, in this sense I want to ask if the authors consider that the inclusion in this study of 13 participants is sufficient and relevant for this research.
- What was the reason why more subjects were not included in this study? I know it was a quarantine period but still such a number of participants is quite difficult to understand.
- Introduction - is well presented and detailed. At line 52 you need to replace the year (2020) with the reference number, also for line 174, 257 etc...
- Lines 80 - 88: ''Introduction should briefly place the study in a broad context and highlight why it is important. It should define the purpose of the work and its significance. The current state of the research field should be carefully reviewed and key publications cited. Please highlight controversial and diverging hypotheses when necessary. Finally, briefly mention the main aim of the work and highlight the principal conclusions. As far as possible, please keep the introduction comprehensible to scientists outside your particular field of research. References should be numbered in order of appearance and indicated by a numeral or numerals in square brackets—e.g., [1] or [2,3], or [4–6]. See the end of the document for further details on references''........what is this?
- After deleting the previous paragraph, please include very clearly, in this chapter, what was the novelty of the study.
- Did the 13 selected subjects present evidence that they are clinically healthy? This is very important, especially since these subjects will perform demanding sports programs for the body.
- Line 106-107- ''Also, the participants completed an electronic version, via Google Forms, of the short-form International Physical Activity Questionnaire (IPAQ).'' What was the reason for completing this questionnaire? the authors do not present any other information regarding this physical activity assessment tool nor any information regarding its content. It is very important for readers to understand very clearly the reason for applying this questionnaire.
- ''Each exercise was followed by 30sec passive recovery''..why did you opt for the passive break? Wasn't it better to take an active break to keep high the excitability of the central nervous system?
- Table 1 - From IPAQ we do not see: high, medium and low PA, where are these variants of physical activity classification? I know that the intensity of an exercise or activity was calculated, but I did not see a PA classification.
- For Table 2 you must add below the table the name for each abbreviation used, also for all the tables, which is the case.
- Line 253-254: ''Several reports have demonstrated that physical activity levels influence HRV adaptations from exercise training [39,40] .''.... in your article we did not see a classification of PA, as we mentioned in the comment above, it was not good to have this information to be able to compare with these reports?
- Line 294-297 - ''should discuss the results and how they can be interpreted from the perspective of previous studies and of the working hypotheses. The findings and their implications should be discussed in the broadest context possible. Future research directions may also be highlighted''....Again, what is this?
- Do you think that the small number of subjects could be a limit of the paper?
Author Response
May 25, 2022
Dear Managing Editor,
On behalf of all the authors, I appreciate the opportunity of obtaining a careful evaluation from the reviewers concerning the article entitled “Short-term high-intensity circuit training does not modify resting heart rate variability in adults during the COVID-19 confinement” (ID: ijerph-1710486). We have answered the questions and the concerns of the reviewers point by point, and the manuscript has been modified according to the suggestions and guidelines of the journal (new information added is marked with blue color), and the revised version is being submitted. The suggestions offered by the reviewers have been especially helpful. My research group and I are sure that the reviewers comments have improved substantially the manuscript content.
Once more, I would appreciate your considering this manuscript in your journal and we hope that this version will be accepted for publication.
Yours faithfully,
Alberto Jiménez Maldonado, Ph.D
Facultad de Deportes Campus Ensenada,
Universidad Autónoma de Baja California,
e-mail: jimenez.alberto86@uabc.edu.mx
Reviewer 1
- The first question for the authors refers to the number of subjects, in this sense I want to ask if the authors consider that the inclusion in this study of 13 participants is sufficient and relevant for this research.
R: We thank the reviewer for this comment. As shown in table 2, the a posteriori total power (1-ß) indicates that the sample was large enough to detect a minimal change (if there were any) in the HRV variables. The computed power corresponds to the probability of detecting a difference equal to or greater than the smallest worthwhile change (threshold) recorded as the relevant difference between the control and the HICT groups for the respective response variable. The very high-power values found (close to 1.0) indicate a very low probability of not rejecting the null hypothesis, given that it is false. Thus, based on the data analyzed, we conclude that the sample size did not explain the absence of significance (i.e., p < 0.05).
- What was the reason why more subjects were not included in this study? I know it was a quarantine period but still such a number of participants is quite difficult to understand.
R: We were unable to recruit more subjects due to prolonged lockdown.
Changes: New information was added in the participants section.
- Introduction - is well presented and detailed. At line 52 you need to replace the year (2020) with the reference number, also for line 174, 257 etc...
R: Thanks for the clarification of our flaw.
Changes: We corrected the references in text.
- Lines 80 - 88: ''Introduction should briefly place the study in a broad context and highlight why it is important. It should define the purpose of the work and its significance. The current state of the research field should be carefully reviewed and key publications cited. Please highlight controversial and diverging hypotheses when necessary. Finally, briefly mention the main aim of the work and highlight the principal conclusions. As far as possible, please keep the introduction comprehensible to scientists outside your particular field of research. References should be numbered in order of appearance and indicated by a numeral or numerals in square brackets—e.g., [1] or [2,3], or [4–6]. See the end of the document for further details on references''........what is this?
R: Thank you for pointing out the mistake.
Changes: We deleted that sentence from the original template.
- After deleting the previous paragraph, please include very clearly, in this chapter, what was the novelty of the study.
R: Thanks for the comment.
Changes: We re-structured the last sentence to mention the novelty of the study.
- Did the 13 selected subjects present evidence that they are clinically healthy? This is very important, especially since these subjects will perform demanding sports programs for the body.
R: Thank you for this observation; the subjects were cleared from any condition that would hindered them from perform high-intense exercises as assessed by the Physical Activity Readiness Questionnaire (PAR-Q).
Changes: We added new information in the experimental procedures section.
- Line 106-107- ''Also, the participants completed an electronic version, via Google Forms, of the short-form International Physical Activity Questionnaire (IPAQ).'' What was the reason for completing this questionnaire? the authors do not present any other information regarding this physical activity assessment tool nor any information regarding its content. It is very important for readers to understand very clearly the reason for applying this questionnaire.
R: Thank you for the comment. The participants completed the IPA-Q to determine the physical activity regime before and after the intervention.
Changes: We added new information in this procedure.
- ''Each exercise was followed by 30sec passive recovery'' Why did you opt for the passive break? Wasn't it better to take an active break to keep high the excitability of the central nervous system?
R: The adapted HICT intervention was modified from the original author (active recovery) because the subjects had to perform 3 exercise sessions per day (36 total in a month), so we decided that a passive recovery will induce a longer time to exhaustion than active recovery https://link.springer.com/article/10.1007/s00421-003-0834-2.
- Table 1 - From IPAQ we do not see: high, medium and low PA, where are these variants of physical activity classification? I know that the intensity of an exercise or activity was calculated, but I did not see a PA classification.
R: Thank you for the comment. The reported average energy expenditure values from the subjects are classified as moderately active individuals.
Changes: We added the cut-off points for physical activity categories in Table 1’s foot.
- For Table 2 you must add below the table the name for each abbreviation used, also for all the tables, which is the case.
R: Thank you for the observation.
Changes: We added the abbreviations names in Table 2.
- Line 253-254: ''Several reports have demonstrated that physical activity levels influence HRV adaptations from exercise training [39,40] .''.... in your article we did not see a classification of PA, as we mentioned in the comment above, it was not good to have this information to be able to compare with these reports?
R: Thank you for the comment.
Changes: We added the classification cut-off points in Table 1.
- Line 294-297 - ''should discuss the results and how they can be interpreted from the perspective of previous studies and of the working hypotheses. The findings and their implications should be discussed in the broadest context possible. Future research directions may also be highlighted''....Again, what is this?
R: Thank you for this comment.
Changes: We deleted that paragraph.
- Do you think that the small number of subjects could be a limit of the paper?
R: Thank you for your comment. We acknowledge that studying large samples will always be desirable over small samples when using a frequentist approach to null hypothesis significance testing, and under normal circumstances. We could have recruited more volunteers for our study; however, the Covid-19 pandemic prevented us from doing it. Large samples do not guarantee that the effects (if any) might be of relevant or practical importance because large samples also elicit trivial statistically significant effects (e.g., p = 0.001) (Levine & Hullett, 2006). Our data suggest that the sample size was not responsible for the lack of statistically significant differences given the a posteriori power analysis.
Levine, T. R., & Hullett, C. R. (2006). Eta Squared, Partial Eta Squared, and Misreporting of Effect Size in Communication Research. Human Communication Research, 28(4), 612-625. https://doi.org/10.1111/j.1468-2958.2002.tb00828.x

Reviewer 2 Report
Review of the manuscript entitled "Short-term high-intensity circuit training does not modify resting heart rate variability in adults during the COVID-19 confinement" for the International Journal of Environmental Research and Public Health.
I highly appreciate the idea of this study. Physical activity is an important part for maintaining during the lockdowns caused by the COVID-19 pandemic. The HICT exercise introduced in this study is a good approach because it is time saving and requires no special gym equipment, so it should be well adhered to. I also appreciate the methodology used in this study. On-line forms and wearable ECG devices with a mobile app were used and such an approach can be used for future development of telemedicine. I appreciate that the experimental part of this study was conducted despite the lockdown and restrictions caused by COVID-19. However, I find some serious shortcomings in the study, especially in the statistical analysis. Please read my comments below.
Line 31: Which kind of stress did you mean? How was the stress assessed? I did not find any psychometric variables related to stress.
Line 69: Yes, HRV can be used to assess the physiological effect as well as psychological stress. How did you distinguish between these sources in this study?
Line 80: This paragraph is a remainder of the template. Remove it.
Line 100: Please specify the randomization procedure. I cannot find any information about it.
Line 108: For devices and software (e.g. Polar H10, Elite HRV, Kubios HRV, R software) the manufacturer, city and country should be indicated. Please check this throughout the manuscript.
Line 118: The figure should be redrawn to comply the requirements of the CONSORT flow diagram. Available at http://www.consort-statement.org/consort-statement/flow-diagram
Line 123: “10 min before” So the participants remained supine for 10 min to stabilize heart rate, followed by another 10 min of recording That is not usual. Rather, participants lay supine for 10 min, with the first 5 min served as stabilization and the last 5 min recorded and analyzed. Please revise this text.
Line 128: Were the RR records visually inspected by an expert and fixed for artifacts if necessary?
Line 130: You used too many interdependent HRV variables. The mean RR is reciprocal to the mean HR, so one of them can be omitted. NN50 and pNN50 are very old indexes of vagal activity and can be omitted, RMSSD alone is sufficient as a time index of vagal activity. LFnu, HFnu, and LF/HF are redundant. Please read https://pubmed.ncbi.nlm.nih.gov/24847279/ Therefore, select and publish only one, preferably LF/HF. Why did not you use a nonlinear index? For example, the sample entropy.
Line 133: Please specify the frequency bands for the calculated HRV frequency indexes.
Line 149: Please specify the calculation of Cohen’s d. I cannot find any information about it. Please note that there are various definitions. Please read https://pubmed.ncbi.nlm.nih.gov/24324449/
Line 152: The ANCOVA used in this study may be appropriate. However, I cannot verify this. Please specify the ANCOVA model using Wilkinson’s notation. Anyway, I mean that ANCOVA is excessive in this case. A repeated measures ANOVA would be sufficient and much more understandable to the readers.
Line 168: The term “technical error of measurement” is not appropriate here because this is not a reliability study. A more appropriate term is “within-subject standard deviation”.
Line 170: Please provide a reference to the definition of smallest worthwhile change SWC = 1.96 × TEM. Note that Buchheit (https://pubmed.ncbi.nlm.nih.gov/24578692/) on page 13 recommended SWC = 0.5 × TEM for HRV indexes. Please justify this issue.
Line 175: HR values should be rounded to integers, more precisely the SD should be rounded to two significant digits. See Hopkins et al (https://pubmed.ncbi.nlm.nih.gov/19092709/), page 5.
Line 175: The values presented here are for HICT, control or combined? If they are combined, this is not good practice in this case. Please provide values separately for HICT and control.
Line 178: There is “min” alone. Is it minutes per day?
Table 1: Please provide the sex distribution. Were they all males or females?
Table 1: This study includes 9 participants in the HICT and 4 in the control group. The sample size of 9 is small and only acceptable in certain circumstances as a costly or time-consuming intervention. It is questionable that this is the case. A sample size of 4 is very small and only acceptable in exceptional circumstances, for example, when laboratory animals are sacrificed or patients with a rare diagnosis. This is not the case in this study. This is the main shortcoming of this study. The statistically insignificant results shown in Table 2 are clearly due to the small sample size. The high power values presented in Table 2 are an illusion because the power is calculated from an unrealistically high SWC = 1.98 TEM. Do you really believe that changes below ~2 TEM are not practically significant? If so, please explain carefully and provide references. In my opinion, you need to include more participants in the control group. If this is not possible, please consider removing the control group and present only the longitudinal changes in the HICT group and name this study as a pilot study.
Figure 2: The whiskers are indistinguishable between S1 and S36. Please explain the symbols “*” and “**”. What statistical test was used?
Table 2: Please provide descriptive statistics for the HRV indexes, specifically mean, standard deviation and Cohen’s d. Also provide the units.
Figure 3: Please describe what black and grey mean. Likely black is the HICT and grey is the control, but it must be clearly presented.
Line 225: I cannot find the resting HR in the results section.
Line 268: HR and RR alone are not indexes of parasympathetic activity. RMSSD is. Please see Buchheit’s review noted above.
Line 273: How was overtraining assessed in this study?
Line 294: Again, a remainder of the template.
Line 318: Please include the Data Availability Statement at the end of the manuscript.
Author Response
May 25, 2022
Dear Managing Editor,
On behalf of all the authors, I appreciate the opportunity of obtaining a careful evaluation from the reviewers concerning the article entitled “Short-term high-intensity circuit training does not modify resting heart rate variability in adults during the COVID-19 confinement” (ID: ijerph-1710486). We have answered the questions and the concerns of the reviewers point by point, and the manuscript has been modified according to the suggestions and guidelines of the journal (new information added is marked with blue color), and the revised version is being submitted. The suggestions offered by the reviewers have been especially helpful. My research group and I are sure that the reviewers comments have improved substantially the manuscript content.
Once more, I would appreciate your considering this manuscript in your journal and we hope that this version will be accepted for publication.
Yours faithfully,
Alberto Jiménez Maldonado, Ph.D
Facultad de Deportes Campus Ensenada,
Universidad Autónoma de Baja California,
e-mail: jimenez.alberto86@uabc.edu.mx
Reviewer 2
I highly appreciate the idea of this study. Physical activity is an important part for maintaining during the lockdowns caused by the COVID-19 pandemic. The HICT exercise introduced in this study is a good approach because it is time saving and requires no special gym equipment, so it should be well adhered to. I also appreciate the methodology used in this study. On-line forms and wearable ECG devices with a mobile app were used and such an approach can be used for future development of telemedicine. I appreciate that the experimental part of this study was conducted despite the lockdown and restrictions caused by COVID-19. However, I find some serious shortcomings in the study, especially in the statistical analysis. Please read my comments below.
- Line 31: Which kind of stress did you mean? How was the stress assessed? I did not find any psychometric variables related to stress.
R: Thank you for this valuable comment
Changes: We re-word that sentence.
- Line 69: Yes, HRV can be used to assess the physiological effect as well as psychological stress. How did you distinguish between these sources in this study?
R: Thank you for your observation in this regard, we agreed with you on how HRV can be utilized for both physiological and psychological variables.
However, we chose to use the HRV as a tool to assess physiological aspects and, in parallel, we applied a questionnaire to attest psychological stress variables (https://opus.lib.uts.edu.au/bitstream/10453/17087/1/2010001392OK.pdf).
Considering the research approach, we chose not to present the questionnaire data and only work with the HRV data in the physiological field.
- Line 80: This paragraph is a remainder of the template. Remove it.
R: Thanks for signaling this mistake.
Changes: We deleted that sentence from the original template.
- Line 100: Please specify the randomization procedure. I cannot find any information about it.
R: Thank you for this observation, the randomization was simple and using an Excel spreadsheet.
Changes: New information was added in the participants section.
- Line 108: For devices and software (e.g. Polar H10, Elite HRV, Kubios HRV, R software) the manufacturer, city and country should be indicated. Please check this throughout the manuscript.
R: Thank you for your comments, the devices manufacturers and cities are seen as bellow:
Polar H10 HR monitor with a Pro Strap (Polar Electro Oy, Kempele, Finland) Kubios Heart Rate Variability standard version software (© 2 2016-2021 Kubios Oy, Finland), the beat-to-beat intervals (R–R) were registered using the free smartphone application Elite HRV® app (Elite HRV LLC, Asheville, NC, USA, Release 4.0.2, 2018). Prism software (GraphPad Inc., San Diego, CA, USA, Release 6.01, 2012). R Core Team. (2021). R: A language and environment for statistical computing. https://www.r-project.org/ and RStudio Team. (2020). RStudio: Integrated Development for R. PBC. http://www.rstudio.com/
Changes: We included this information in the methods section.
- Line 118: The figure should be redrawn to comply the requirements of the CONSORT flow diagram. Available at http://www.consort-statement.org/consort-statement/flow-diagram
R: Thanks for your valuable suggestion
Changes: We modified the former format to the CONSORT format.
- Line 123: “10 min before” So the participants remained supine for 10 min to stabilize heart rate, followed by another 10 min of recording That is not usual. Rather, participants lay supine for 10 min, with the first 5 min served as stabilization and the last 5 min recorded and analyzed. Please revise this text.
R: We employed the methods suggested for perturbations on the ANS, due to the uncertainty of getting the baseline data disrupted by the individual’s state during lockdown. Furthermore, following the Task Force of the European Society of Cardiology and the North American Society of Pacing Electrophysiology guidelines, a display of stacked series of sequential power spectra (for example, over 20 min) may help confirm steady state conditions for a given physiological state. https://doi.org/10.1161/01.CIR.93.5.1043
- Line 128: Were the RR records visually inspected by an expert and fixed for artifacts if necessary?
R: Thanks for your observation, the RR recordings were analyzed by the first author of the MS, and the artifacts were fixed at very low correction.
Changes: We corrected the latter sentence in HRV analysis.
- Line 130: You used too many interdependent HRV variables. The mean RR is reciprocal to the mean HR, so one of them can be omitte NN50 and pNN50 are very old indexes of vagal activity and can be omitted, RMSSD alone is sufficient as a time index of vagal activity.LFnu, HFnu, and LF/HF are redundant. Please read https://pubmed.ncbi.nlm.nih.gov/24847279/ Therefore, select and publish only one, preferably LF/HF. Why did not you use a nonlinear index? For example, the sample entropy.
R: Thank you for your comment, even though these time domain variables are redundant between each other, the report was made in order to be descriptive and inferential for other cardiac performance characteristics.
For non-linear indexes, it was previously reported that SD1 and SD2 are the equivalents for RMSSD, HF and LF respectively https://doi.org/10.1002/mus.25573 and for sample entropy, it requires at least 800 RR intervals to be analyzed 10.1007/s10877-009-9210-z.
- Line 133: Please specify the frequency bands for the calculated HRV frequency indexes.
R: Thanks for your observation. We will specify the frequency bands in the MS.
Changes: New information was included in the HRV analysis section.
- Line 149: Please specify the calculation of Cohen’s d. I cannot find any information about it. Please note that there are various definitions. Please read https://pubmed.ncbi.nlm.nih.gov/24324449/
R: Thank you for this comment.
Changes: We added our Cohen’s d values and its calculation as stated by Cohen, 1992.
- Line 152: The ANCOVA used in this study may be appropriate. However, I cannot verify this. Please specify the ANCOVA model using Wilkinson’s notation. Anyway, I mean that ANCOVA is excessive in this case. A repeated measures ANOVA would be sufficient and much more understandable to the readers.
R: Thank you for this comment. We sought statistical advice, and we discussed the strengths and limitations of ANOVA for this specific data set. We decided to include ANCOVA to remove the initial error variance in both, the control and the experimental groups. We believe that this is a powerful approach for analyzing our data. Each data obtained after the training period was used as a response variable to build a regression using as a design factor the subjects' grouping variable (i.e., control, exercise) and, as a covariate, the baseline measurements. The regression model built used the concept of the sum of squares of marginal regression to analyze the variability that explains whether performing HICT or not once the corresponding covariate has been included, and that has previously explained a part of the total sum of squares.
- Line 168: The term “technical error of measurement” is not appropriate here because this is not a reliability study. A more appropriate term is “within-subject standard deviation”.
R: Thank you for your observation. We have changed the term accordingly.
- Line 170: Please provide a reference to the definition of smallest worthwhile change SWC = 1.96 × TEM. Note that Buchheit (https://pubmed.ncbi.nlm.nih.gov/24578692/) on page 13 recommended SWC = 0.5 × TEM for HRV indexes. Please justify this issue.
R: Thank you for this comment. We agreed with the reviewer, and based on your suggestion, we have computed the new SWCs according to Buchheit’s paper, which seems more appropriate for this study than the approach we initially followed. Consequently, the new SWC data are reported on Table 2.
Buchheit M. (2014). Monitoring training status with HR measures: do all roads lead to Rome?. Frontiers in physiology, 5, 73. https://doi.org/10.3389/fphys.2014.00073
- Line 175: HR values should be rounded to integers, more precisely the SD should be rounded to two significant digits. See Hopkins et al (https://pubmed.ncbi.nlm.nih.gov/19092709/), page 5.
R: Thank you for your observation.
Changes: Adjust the HR values to integers
- Line 175: The values presented here are for HICT, control or combined? If they are combined, this is not good practice in this case. Please provide values separately for HICT and control.
R: Thank you for your comment, the values of HR reported in this line were exclusively done for the HICT, for the purpose of showing the exercise intensity for this specific group.
Changes: We re-structure the sentence and now is clear to which group is referred the HR values.
- Line 178: There is “min” alone. Is it minutes per day?
R: We thank the reviewer for this question, it is minutes per day.
Changes: We corrected that line.
- Table 1: Please provide the sex distribution. Were they all males or females?
R: We thank the reviewer for this suggestion.
Changes: We added sex distribution in table 1.
- Table 1: This study includes 9 participants in the HICT and 4 in the control group. The sample size of 9 is small and only acceptable in certain circumstances as a costly or time-consuming intervention. It is questionable that this is the case. A sample size of 4 is very small and only acceptable in exceptional circumstances, for example, when laboratory animals are sacrificed or patients with a rare diagnosis. This is not the case in this study. This is the main shortcoming of this study. The statistically insignificant results shown in Table 2 are clearly due to the small sample size. The high power values presented in Table 2 are an illusion because the power is calculated from an unrealistically high SWC = 1.98 TEM. Do you really believe that changes below ~2 TEM are not practically significant? If so, please explain carefully and provide references. In my opinion, you need to include more participants in the control group. If this is not possible, please consider removing the control group and present only the longitudinal changes in the HICT group and name this study as a pilot study.
R: Thank you for your comment. We acknowledge that studying large samples will always be desirable over small samples when using a frequentist approach to null hypothesis significance testing, and under normal circumstances. We could have recruited more volunteers for our study; however, the Covid-19 pandemic prevented us from doing it. Large samples do not guarantee that the effects (if any) might be of relevant or practical importance because large samples also elicit trivial statistically significant effects (e.g., p = 0.001) (Levine & Hullett, 2006). Our data suggest that the sample size was not responsible for the lack of statistically significant differences given the a posteriori power analysis. At this time, we do not know whether the new SWC values computed according to Bucheit (2014) have a clinical or practical importance.
Buchheit M. (2014). Monitoring training status with HR measures: do all roads lead to Rome?. Frontiers in physiology, 5, 73. https://doi.org/10.3389/fphys.2014.00073
Levine, T. R., & Hullett, C. R. (2006). Eta Squared, Partial Eta Squared, and Misreporting of Effect Size in Communication Research. Human Communication Research, 28(4), 612-625. https://doi.org/10.1111/j.1468-2958.2002.tb00828.x
- Figure 2: The whiskers are indistinguishable between S1 and S36. Please explain the symbols “*” and “**”. What statistical test was used?
R: Thank you for indicating this error on the figure legend.
Changes: We corrected the legend with new information.
- Table 2: Please provide descriptive statistics for the HRV indexes, specifically mean, standard deviation and Cohen’s d. Also provide the units.
R: Thank you for your comment
Changes: We added in the table 2 descriptive statistics for the HRV indexes (mean, standard deviation and Cohen’s d).
- Figure 3: Please describe what black and grey mean. Likely black is the HICT and grey is the control, but it must be clearly presented.
R: Thank you for your comment
Changes: We added this group depiction on the figure legend.
- Line 225: I cannot find the resting HR in the results section.
R: Thank you for your comment
Changes: The resting HR data have been included in the table 2.
- Line 268: HR and RR alone are not indexes of parasympathetic activity. RMSSD is. Please see Buchheit’s review noted above.
R: Thank you for your comment.
Changes: we re-structure that sentence.
- Line 273: How was overtraining assessed in this study?
R: Thank you for your comment. We use the same HRV indexes to monitor for any physiological overreaching element in the HICT group https://doi.org/10.1046/j.1475-0961.2003.00523.x
Changes: We re-structured that line.
- Line 294: Again, a remainder of the template.
R: Thank you for your observation.
Changes: We deleted that paragraph
- Line 318: Please include the Data Availability Statement at the end of the manuscript.
R: Thanks for the suggestion
Changes: Data Availability Statement was included before Acknowledgements.

Reviewer 3 Report
Thank you for the opportunity to review this manuscript, which considers some interesting issues. However, there are some significant flaws that must be addressed. Please see the comments.
Introduction
Line 79 is [20] [ 21], should be [20,21]
Lines 80-88: I guess these are the suggestions of the previous reviewer. Maybe it is better to remove them from the main text.
Material and Methods
There is no doubt that such research in the pandemic era is important. The small sample size is lowering the impact of the presented results. Although the authors indicated in the limitation paragraph why the sample is so small, still the 4 participants in the control group are not enough to draw conclusions
Line 107: How the survey results have been converted into MET’s unit? It is worthwhile to complete this.
The HICT took about 12 minutes and the time of vigorous activity was between 27.5 and 47.5 minutes. Did the HICT was additional exercise or was it completed during the general time of vigorous activity?
Did the control group was allowed to perform any vigorous activity during the experiment? What the daily time of this activity was?
Discussion
Lines 272-273: “…it demonstrates that HICT is an exercise modality that does not induce overtraining (including stress and irritability)…” how the stress and irritability were estimated?
Lines 294-297: I guess these are the suggestions of the previous reviewer. Maybe it is better to remove them from the main text.
Author Response
May 25, 2022
Dear Managing Editor,
On behalf of all the authors, I appreciate the opportunity of obtaining a careful evaluation from the reviewers concerning the article entitled “Short-term high-intensity circuit training does not modify resting heart rate variability in adults during the COVID-19 confinement” (ID: ijerph-1710486). We have answered the questions and the concerns of the reviewers point by point, and the manuscript has been modified according to the suggestions and guidelines of the journal (new information added is marked with blue color), and the revised version is being submitted. The suggestions offered by the reviewers have been especially helpful. My research group and I are sure that the reviewers comments have improved substantially the manuscript content.
Once more, I would appreciate your considering this manuscript in your journal and we hope that this version will be accepted for publication.
Yours faithfully,
Alberto Jiménez Maldonado, Ph.D
Facultad de Deportes Campus Ensenada,
Universidad Autónoma de Baja California,
e-mail: jimenez.alberto86@uabc.edu.mx
Reviewer 3
Thank you for the opportunity to review this manuscript, which considers some interesting issues. However, there are some significant flaws that must be addressed. Please see the comments.
Introduction
- Line 79 is [20] [ 21], should be [20,21]
Changes: We corrected that mistake
- Lines 80-88: I guess these are the suggestions of the previous reviewer. Maybe it is better to remove them from the main text.
Changes: We deleted that paragraph
Material and Methods
- There is no doubt that such research in the pandemic era is important. The small sample size is lowering the impact of the presented results. Although the authors indicated in the limitation paragraph why the sample is so small, still the 4 participants in the control group are not enough to draw conclusions
R: We thank the reviewer for this comment. As shown in table 2, the a posteriori total power (1-ß) indicates that the sample was large enough to detect a minimal change (if there were any) in the HRV variables. The very high-power values found (close to 1.0) indicate that the absence of statistical significance (i.e., p < 0.05) was not explained by the sample size. Nevertheless, we acknowledge that studying large samples is desirable over small samples under normal circumstances. As mentioned in the manuscript, we could have recruited more volunteers for our study; however, the Covid-19 pandemic prevented us from doing it. In addition, large samples do not guarantee that the effects (if any) might be of relevant or practical importance because large samples also elicit trivial statistically significant effects (e.g., p = 0.001) (Levine & Hullett, 2006).
Levine, T. R., & Hullett, C. R. (2006). Eta Squared, Partial Eta Squared, and Misreporting of Effect Size in Communication Research. Human Communication Research, 28(4), 612-625. https://doi.org/10.1111/j.1468-2958.2002.tb00828.x
- Line 107: How the survey results have been converted into MET’s unit? It is worthwhile to complete this.
Changes: We added new information in methods and Table 1 for MET’s calculation and cut-off values for the subjects’ categorization.
- The HICT took about 12 minutes and the time of vigorous activity was between 27.5 and 47.5 minutes. Did the HICT was additional exercise or was it completed during the general time of vigorous activity?
R: The time reported for the HICT was in total minutes per day. Since the subjects performed the 3 sessions a day, 3 times a day for 4 weeks.
- Did the control group was allowed to perform any vigorous activity during the experiment? What the daily time of this activity was?
R: Thanks for your question, unfortunately, the information vigorous activity (minutes per day) was missed, the researches employed Google Forms questionnaires for the collection of IPAQ data at post-intervention, however, the CON did not answer it. Nevertheless, we considered that the CON maintained the daily life-style, a similar pattern showed at the pre intervention. The reason of this hypothesis is that the work was performed during the first wave of the lockdown (same period).
Discussion
- Lines 272-273: “…it demonstrates that HICT is an exercise modality that does not induce overtraining (including stress and irritability)…” how the stress and irritability were estimated?
R: Thank you very mucho for this interesting observation
Changes: We re-structure that sentence.
- Lines 294-297: I guess these are the suggestions of the previous reviewer. Maybe it is better to remove them from the main text.
Changes: We removed that part from the text

Round 2
Reviewer 1 Report
Line 113-117: ''The participants also completed an electronic version, of the short-form International Physical Activity Questionnaire (IPAQ) to describe the physical activity habits before and after HICT. Additionally, the Physical Activity Readiness Questionnaire (PAR-Q) was applied in order to assess the participants’ health status to perform highly demanding exercises via Google Forms''.....Please add information about the IPAQ-SF questionnaire, even if it is a secondary application tool it is very important to provide information about the questionnaire.
Rearrange the Table number 2 on the page, it looks very mess and the values obtained are no longer understood.
Author Response
Dear Managing Editor,
On behalf of all the authors, I appreciate the opportunity of obtaining a careful evaluation from the reviewers concerning the article entitled “Short-term high-intensity circuit training does not modify resting heart rate variability in adults during the COVID-19 confinement” (ID: ijerph-1710486). We have answered the questions and the concerns of the reviewers point by point, and the manuscript has been modified according to the suggestions and guidelines of the journal (new information added is marked with blue color), and the revised version is being submitted. The suggestions offered by the reviewers have been especially helpful. My research group and I are sure that the reviewers comments have improved substantially the manuscript content.
Once more, I would appreciate your considering this manuscript in your journal and we hope that this version will be accepted for publication.
Yours faithfully,
Alberto Jiménez Maldonado, Ph.D
Facultad de Deportes Campus Ensenada,
Universidad Autónoma de Baja California,
e-mail: jimenez.alberto86@uabc.edu.mx
Reviewer 1
- Line 113-117: ''The participants also completed an electronic version, of the short-form International Physical Activity Questionnaire (IPAQ) to describe the physical activity habits before and after HICT. Additionally, the Physical Activity Readiness Questionnaire (PAR-Q) was applied in order to assess the participants’ health status to perform highly demanding exercises via Google Forms''.....Please add information about the IPAQ-SF questionnaire, even if it is a secondary application tool it is very important to provide information about the questionnaire.
R: Thanks to the reviewer for this comment in order to improve the text.
Change: New information has been added to detail in Experimental procedures section.
- Rearrange the Table number 2 on the page, it looks very mess and the values obtained are no longer understood.
R: Thanks to the reviewer for this comment.
Changes: The table 2 has been edited to clarify the data presented.

Reviewer 2 Report
The manuscript has been improved and I am satisfied with author’s responses. I have one request that can be addressed when copyediting the manuscript. The mean ± SD and SWC values in Table 2 should be more appropriately rounded. Three places after the decimal point are too many. One place after the decimal point is sufficient.
Author Response
Dear Managing Editor,
On behalf of all the authors, I appreciate the opportunity of obtaining a careful evaluation from the reviewers concerning the article entitled “Short-term high-intensity circuit training does not modify resting heart rate variability in adults during the COVID-19 confinement” (ID: ijerph-1710486). We have answered the questions and the concerns of the reviewers point by point, and the manuscript has been modified according to the suggestions and guidelines of the journal (new information added is marked with blue color), and the revised version is being submitted. The suggestions offered by the reviewers have been especially helpful. My research group and I are sure that the reviewers comments have improved substantially the manuscript content.
Once more, I would appreciate your considering this manuscript in your journal and we hope that this version will be accepted for publication.
Yours faithfully,
Alberto Jiménez Maldonado, Ph.D
Facultad de Deportes Campus Ensenada,
Universidad Autónoma de Baja California,
e-mail: jimenez.alberto86@uabc.edu.mx
Reviewer 2
The manuscript has been improved and I am satisfied with author’s responses. I have one request that can be addressed when copyediting the manuscript. The mean ± SD and SWC values in Table 2 should be more appropriately rounded. Three places after the decimal point are too many. One place after the decimal point is sufficient.
R: Thanks to the reviewer for this comment.
Changes: The table 2 has been edited to clarify the data presented.
